# Effects of Valine and Urea on Carbon and Nitrogen Accumulation and Lignin Content in Peach Trees

**DOI:** 10.3390/plants12081596

**Published:** 2023-04-10

**Authors:** Maoxiang Sun, Suhong Li, Haixiang Yu, Qingtao Gong, Binbin Zhang, Guangyuan Liu, Yuansong Xiao, Futian Peng

**Affiliations:** 1State Key Laboratory of Crop Biology, College of Horticulture Science and Engineering, Shandong Agricultural University, Tai’an 271018, China; 2Shandong Institute of Pomology, Tai’an 271018, China

**Keywords:** valine, new shoots, lignin, ammonium nitrogen, nitrate nitrogen, carbon and nitrogen metabolism

## Abstract

Nitrogen availability and uptake levels can affect nutrient accumulation in plants. In this study, the effects of valine and urea supplementation on the growth of new shoots, lignin content, and carbon and the nitrogen metabolism of ‘*Ruiguang 39/peach*’ were investigated. Relative to fertilization with urea, the application of valine inhibited shoot longitudinal growth, reduced the number of secondary shoots in autumn, and increased the degree of shoot lignification. The application of valine also increased the protein level of sucrose synthase (SS) and sucrose phosphate synthase (SPS) in plant leaves, phloem, and xylem, thereby increasing the soluble sugar and starch content. It also resulted in an increase in nitrate reductase (NR), glutamine synthase (GS), and glutamate synthase (GOGAT) protein levels, with an increase in plant contents of ammonium nitrogen, nitrate nitrogen, and soluble proteins. Although urea application increased the protein level of carbon- and nitrogen-metabolizing enzymes, the increase in plant growth reduced the overall nutrient accumulation and lignin content per unit tree mass. In conclusion, the application of valine has a positive effect on increasing the accumulation of carbon and nitrogen nutrients in peach trees and increasing the lignin content.

## 1. Introduction

Peach trees grow and develop rapidly, with a high natural pollination and fruit setting rate, wide adaptability, and high cultivation value. As one of the main peach-producing areas, China ranks first in the world in terms of yield and area [1]. The peach tree belongs to Arbor; however, the peach tree grows too luxuriant and tall, and brings great difficulties to the fruit farmer when it is time for harvest [2]. Therefore, the regulation of tree height growth is particularly important. Nitrogen, as the main component of proteins, nucleic acids, and lipids, is one of the essential mineral nutrients for the growth of peach trees. In agriculture, urea is used extensively as a nitrogen fertilizer to promote crop growth. Urea nitrogen enters the plant either directly or in the form of ammonium or nitrate after urea degradation by soil microbes [3]. Gu (1981) divided the nitrogen demand period of apples into three stages, where the last stage is after fruit harvesting and is called the nitrogen storage period [4]. Both peach and apple trees belong to the group of northern deciduous fruit trees. They lose their leaves in autumn and grow new leaves in the following spring. The nutrients needed for overwintering and flowering in the following year come from the tree body reserves. The regulation of plant carbon and nitrogen metabolism, as well as the relationship between vegetative growth and nutrient accumulation, has an important impact on the quality of flower buds and the rate of fruit setting [4]. Carbon and nitrogen metabolism is one of the basic physiological metabolic activities of plants [5]. The intensities of carbon and nitrogen metabolism and their interaction are important for maintaining the balance between vegetative and reproductive growth, as well as nutrient accumulation and transformation [6], thereby affecting the yield and quality of the fruit [7].

Carbon metabolism includes processes such as light energy fixation, the synthesis of energy substances, and the conversion and accumulation of carbohydrates [8]. Sucrose synthase (SS) and sucrose phosphate synthase (SPS) play key roles in carbon metabolism [9,10]. Plants cannot directly utilize inorganic nitrogen, which can only be absorbed by plants after assimilation into organic nitrogen through nitrogen metabolism. Nitrate reductase (NR), glutamate synthase (GOGAT), and glutamine synthase (GS) play key roles in nitrogen metabolism [11]. Lignin is formed via the oxidative polymerization of p-hydroxycinnamyl alcohol monolignols coming from the phenylpropanoid pathway and differing in the number of methoxyl groups on the aromatic ring [12]. This polyphenolic compound is essential for the growth of land plants because it provides mechanical support to the whole plant and is responsible for the hydrophobic proprieties of water-conducting cells in the xylem [13].

Considering the increasing demand for peach fruit and the labor and raw material costs associated with the expansion of the peach tree planting area [14], producers have turned to the use of various plant growth regulators to increase fruit yields as a more cost-effective measure. However, most of these are synthetic, classified as pesticides [15], and remain within fruits and soil [16], which affects human and animal health, so many countries and regions regulate the maximum residuals which can remain after their use. At present, amino acid fertilizers are mainly produced by the fermentation and concentration of plants, animal hair, industrial wastewater, etc., and have the advantages of being green, safe, and favoring environmental protection. Amino acids are not only a nitrogen source that can be directly absorbed by plants, but are also the major transport form of organic nitrogen in plants [17]. Amino acids have been reported to promote the growth of animals and plants [18,19,20,21,22], such as glycine, which enhances the enzymatic protection system and promotes carbon and nitrogen metabolism to accelerate the growth of tobacco plants and improve the quality of tobacco leaves [23]. Cao’s research (2012) found that compared with the sole use of nitrate fertilizer, a 20% replacement of the nitrates with amino acids significantly reduced the nitrate content in pakchoi leaves, with an increase in the contents of soluble proteins, total nitrogen, soluble sugars, and free amino acids [24]. In addition, this replacement significantly increased the content of ammonium nitrogen, free amino acids, and soluble protein in root exudates. Valine has been reported to inhibit the growth of new shoots of peach trees in an environmentally friendly manner, by regulating the balance of *PpSnRK1* (sucrose non-fermenting-1-related protein kinase), *PpTOR* (Target of Rapamycin), and the synthesis of isoleucine [25].

Previous studies have found that valine inhibited peach tree new shoot growth, shortened internode length, and had no adverse effects on the stem diameter or leaf and fruit quality. However, there have been no reported studies on the regulation of carbon and nitrogen metabolism in peach trees by valine, or of its effects on the growth of new shoots and lignin accumulation. In this study, two-year-old ‘*Ruiguang 39*’ peach trees were used as materials to investigate the effects of valine on the metabolism of carbon and nitrogen and on the growth and nutrient accumulation of new shoots.

## 2. Results

### 2.1. Effects of Valine and Urea on the Growth of New Shoots of Peach Trees

Figure 1A shows the distribution of the main branches and side branches of new shoots of peach trees. As can be seen from Figure 1B, the number of new secondary autumn shoots increased significantly compared with the control after valine or urea were applied. The application of valine increased lateral branch number (27%; Figure 1C), lateral branch length (38%; Figure 1D), and fresh weight (10%; Figure 1F) compared with the control treatment. Relative to the urea treatment, the application of valine showed significant reductions (*p* < 0.05) in the number of lateral branches (−34%; Figure 1C) and the growth lengths of the lateral branch (−22.1%; Figure 1D) and main branch (−23.2%; Figure 1E), resulting in a significant reduction in plant biomass fresh weight (−12.8%; Figure 1F).

### 2.2. Effects of Valine and Urea on Lignin Content in New Shoots of Peach Trees

It can be seen in Figure 2A that the valine treatment resulted in a more intense phloroglucinol staining in sections of new shoots, indicating a higher lignin content relative to that in shoots after the urea treatment. It can be seen from Figure 2B that the different treatments have different effects on the color of new shoots. Compared with the control, the valine treatment resulted in redder shoots, while shoots of urea-fertilized plants were greener. Urea application slightly but significantly (*p* < 0.05) increased the average shoot diameter (6.4%, Figure 2C), with significant decreases in puncture strength (7.8%; Figure 2D) and lignin content (9.9%; Figure 2E).

### 2.3. The Effect of Valine and Urea on Carbon Metabolism Enzymes in Peach Tree

As can be seen from Figure 3A, the SS protein level of leaves in the valine treatment group was the highest at 15 days, then gradually decreased to return to control levels at 60 days. As can be seen from Figure 3B, the SS protein level in the phloem was consistently and significantly higher than that of the control over the test period, whereas the SS protein levels in xylem were relatively and significantly higher than the control between 15 and 45 days, but insignificantly different from the control at 60 days (Figure 3C). The SS enzyme protein level of leaves in urea treatment was higher than that in valine treatment only at 60 d and 15 d in phloem.

Figure 3D,E,F shows that the SPS protein levels of the urea treatment group were significantly lower than the control levels in leaves at 45 days and 60 days, in phloem at 60 days, and in xylem at 15 days. However, both leaf SPS protein level and phloem SPS protein level under valine treatment were significantly higher than those under urea treatment at all time points, and phloem SPS protein level was significantly higher than that under urea treatment at 15, 30, and 45 days (Figure 3D–F).

### 2.4. Effects of Valine and Urea on Carbon Metabolites in Peach Tree

Figure 4A shows that the shape and area of stomata in peach leaves were different after urea and valine treatment. Leaves treated with valine displayed full stomata and a larger stomata area, which contributed to gas exchange between cells and the outside environment (Table 1). It can be seen from Figure 4B,C that the net photosynthetic rate and chlorophyll content in the valine treatment group were higher than in the control and urea treatments after 15 days and at 60 days. Figure 4D displays the soluble sugar contents 60 d after treatment, where the soluble sugar content in leaves, compared with control phloem and xylem of the valine treatment group, was significantly increased by 19.1%, 18.7%, and 32.6%, respectively. Conversely, the soluble sugar content of urea treatment was significantly lower than that of valine treatment in different tissues.

From Figure 4E, it can be seen that both valine and urea treatments significantly increased the starch content in leaves, whereas the starch content of xylem and phloem in valine treatment was significantly higher than that in urea treatment.

### 2.5. Effects of Valine and Urea on Enzymes of Nitrogen Metabolism in Peach Tree

It can be seen from Figure 5A that relative to the control, the valine treatment consistently increased the NR activity of leaves over the entire test period, whereas the effects of the urea treatment appeared to be limited to 15–30 days. The leaf NR enzyme activity of the valine-treated leaves was 25.4% and 30.7% higher than that of the urea treatment and control at 60 days, respectively. Conversely, relative to the control, the valine treatment had less effect on phloem and xylem NR activities than the urea treatment, which was more obvious in the 15–45 day period (Figure 5B,C).

It can be seen from Figure 5D that on the 30th day to 60th day, valine treatment made the GS protein level in leaves higher than the other two groups, and it shows a trend of first decreasing and then increasing. On the 30th day, the GS protein level of valine-treated leaves was 16.7% and 42.9% higher than that of urea and control, respectively. The valine treatment had greater relative effects on phloem and xylem GS protein levels in the later stage of treatment, while the treatment with urea had a greater effect on phloem and xylem GS protein levels in the early stages of treatment (Figure 5E,F).

It can be seen from Figure 5G,H, and I that both valine and urea treatments have a promoting effect on GOGAT protein level in leaves, phloem, and xylem. The GOGAT enzyme protein level of leaves from the valine treatment group remained higher than that of the control and urea treatment group from 30 days. The GOGAT enzyme protein level in the phloem of valine-treated plants was also higher than that of the urea or control groups from the 45–60-day period. Conversely, the protein level of xylem GOGAT in valine-treated plants was similar to that after urea treatment, with protein level being higher than that of the control.

### 2.6. Effects of Valine and Urea on Nitrogen Metabolites in Peach Tree

After both valine and urea treatment, the contents of nitrate (Figure 6A) and ammonium forms of nitrogen (Figure 6B) in leaves, phloem, and xylem were increased. However, in the urea treatment group, the contents of ammonium nitrogen in the three tissues were significantly higher than those in the valine treatment group, indicating that the effect of urea on the ammonium nitrogen content of plants was greater than that of valine.

As shown in Figure 6C, the total nitrogen content in leaves after valine treatment was significantly higher than that of the control or urea-treated plants, and both the valine and urea treatments resulted in higher phloem contents of total nitrogen relative to the control.

Figure 6D shows that valine and urea treatments had no significant effect on the soluble protein content of leaves. However, relative to the control, the treatment of the plants with urea resulted in a decreased soluble protein content in both phloem and xylem, whereas the soluble protein contents after the valine treatment were increased in the phloem and the xylem.

### 2.7. Correlation Effects of Different Treatments on Key Enzymes, Carbon and Nitrogen Metabolism and Accumulation

A membership function is a curve that maps input data points into values between 0 and 1, representing their degree of membership. In this study, the greater the membership function value, the stronger the relationship between this index and carbon and nitrogen metabolism. Membership function analysis of SS, SPS, NR, GOGAT, and GS in the leaves, phloem, and xylem of each treatment group was performed. In Table 2, it can be seen that the comprehensive effects of each treatment on the key enzymes of carbon and nitrogen metabolism in plant leaves, phloem, and xylem are in the following order: valine > urea > control.

Membership function analysis was performed on the contents of soluble sugar, starch, nitrate nitrogen, ammonium nitrogen, and soluble protein in each treatment group (Table 3). According to the principle that the larger the mean value, the higher the nutrient content per unit mass, it can be seen that the nutrient content per unit mass in the valine treatment group is the highest, followed by the urea treatment and the control.

Figure 7 exhibits a heatmap correlation matrix among different physiological traits. For the correlation study, the lignin content of the peach tree as well as carbon metabolism enzyme protein level (SS and SPS) and nitrogen metabolism NR enzyme activity (GS and GOGAT enzyme protein level) were evaluated. As can be seen from Figure 7, the lignin contents in peach tree were positively correlated with the plant nitrogen metabolism enzyme protein level. The net photosynthetic rate in peach tree was positively correlated with plant SS protein level, the chlorophyll content, the starch content, and SPS protein level, and the new shoot diameters correlation was significantly different and was negatively correlated with the soluble proteins and the puncture strength.

## 3. Discussion

Lignification is one of the final stages of xylem cell differentiation, where lignin is deposited within the carbohydrate matrix of the cell wall by the infilling of interlamellar voids, and at the same time, the formation of chemical bonds with the non-cellulosic carbohydrates [26,27]. The lignification and lignin content of plants are closely related to their growth and development. Our study found that valine can enhance the lignification of peach trees; compared with urea, the puncture strength, lignin content, and phloroglucinol staining depth were significantly increased after valine treatment, indicating that the application of valine can increase the degree of lignification of peach shoots. Therefore, the application of valine can control the growth height of peach trees, enhance the convenience of fruit harvest, and play an important role in peach production.

As two basic metabolic activities in plants, carbon and nitrogen metabolism are dynamically balanced in plants and are interdependent. Carbon metabolism can provide the carbon source and energy required for nitrogen metabolism, and nitrogen metabolism can provide feedstock and various enzymes and proteins for carbon metabolism. The level of nitrogen supply and uptake not only affects the nutrient accumulation of plants [28], but also affects the resistance of plants to biotic and abiotic stresses [29,30]. SS and SPS are two important enzymes in the process of plant carbon metabolism. Studies have shown that in maize leaves, the protein levels of SS and SPS are related to the photosynthetic rate and the conversion of sucrose [31]. Hu et al. showed that the application of different proportions of ammonium nitrogen and amino acid nitrogen in different periods could significantly change the growth characteristics and carbon metabolism activities of sugar beet [32]. In this study, the leaf SPS protein level began to decrease at 30 days after urea treatment and was significantly lower than that of the control group after 45 and 60 days, while the net photosynthetic rate did not decrease. The SS protein level in leaves after valine treatment was significantly higher than that in the control from 15 to 60 days, while the SPS protein level increased from 15 to 30 days and then decreased. These results suggested that valine treatment could improve carbon accumulation. Stomata are the channels by which plants exchange water vapor and carbon dioxide with the environment. Valine treatment can increase the net photosynthetic rate and chlorophyll content of leaves, thereby increasing gas exchange with the external environment [33]. The protein levels of SS and SPS both affect the content of total soluble sugar and starch [34]. The soluble sugar contents in leaves, phloem, and xylem were all significantly increased after valine treatment, but there was no difference between the urea treatment and the control. The soluble protein contents in phloem and xylem also decreased significantly after urea treatment, which may be due the effect of urea supplements on the mobilization of plant nutrients towards vegetative growth, with a concomitant reduction in nutrient reserves. Compared with urea treatment, the starch content of leaves increased after valine treatment [35].

Nitrogen is absorbed in the plant mainly from ammonium and nitrate forms of nitrogen and their contents in the plant are relatively stable. NR is a key enzyme in nitrogen metabolism. NR and GS are the two rate-limiting enzymes that regulate the absorption and transformation of ammonium and nitrate ions in plants [36]. At present, soil application of urea is a common form of nitrogen supplementation. The application of urea to soybean can improve the protein level of these enzymes and accelerate ammonium and nitrate ion absorption [37]. Compared with inorganic nitrogen application, Phe as the only nitrogen source in poplar significantly promoted root growth, reduced inorganic nitrogen absorption and assimilation, enhanced organic nitrogen metabolism, and improved nitrogen utilization efficiency [38]. Our research found that the NR activity of leaves in the urea and valine treatment groups was higher than or equal to that in the control group, but the effect on leaves was greater than that in phloem and xylem. The effects of valine on NR activity were sustained longer than those of urea. GS and GOGAT were also significantly increased in the two treatment groups or were maintained at the same level as the control group, and only urea treatment reduced the protein level of GOGAT in xylem. Previous studies have also shown that the use of nitrogen-based fertilizers significantly increases the plant’s ammonium and nitrate contents relative to plant grown without fertilizer [39], which is consistent with the findings in this report.

Through membership function analysis, we found that valine treatment had the highest effect on the key enzymes of carbon and nitrogen metabolism in plant leaves, phloem, and xylem (Table 2). Membership function analysis was also performed on the contents of soluble sugar, starch, nitrate nitrogen, ammonium nitrogen, and soluble protein in each treatment group. We found that the peach tree treated with valine had the highest nutrient content per unit mass (Table 3). The membership function analysis showed that fertilization with valine can promote the accumulation of carbon and nitrogen in peach trees. We found through correlation analysis that lignin content is closely related to nitrogen metabolism. However, puncture strength was strongly correlated with carbon metabolism. (Figure 7). Therefore, the application of valine plays an important role in the accumulation of lignin and the control of the growth of peach shoots.

## 4. Materials and Methods

### 4.1. Experimental Materials and Experimental Design

The experiment was carried out in the experimental base of Shandong Agricultural University (Tai’an, China) from July to September 2020. The growth conditions of the test materials were the same, and the basic physical and chemical properties of the tested soil were as follows: pH value was 6.53, alkaline hydrolyzable nitrogen content was 47.53 mg∙kg^−1^, organic matter content was 13.47 g∙kg^−1^, available phosphorus content was 38.65 mg∙kg^−1^, and the available potassium content was 87.33 mg∙kg^−1^. The annual average temperature is 13 °C, and the highest temperature was 26.4 °C in July, and the lowest in January, with an average of 2.6 °C. The average annual precipitation is 697 mm. The rootstock used was mountain peach Prunus davidiana. In total, 30 peach trees of ‘*Ruiguang 39*’, with a spacing of 2 m × 5 m, were used as the test materials, and were routinely managed during the test. Pre-experiment found that the valine concentration of 4 g·L^−1^ was the most effective, so it was set as experimental concentration (Table 4). The following three treatments were employed: 2 L water (control), 2 L valine solution with a nitrogen content of 4 g·L^−1^ (valine), and 2 L urea solution with a nitrogen content of 4 g·L^−1^ (urea). The treatments were applied once by pouring, at eight in the morning. Ten trees were selected for each treatment and three biological replicates were performed.

Samples were taken at 0, 15, 30, 45, and 60 days after treatment to determine the activities of enzymes related to carbon and nitrogen metabolism. Our research found that there was no significant difference in the protein level of carbon- and nitrogen-metabolism-related enzymes on day 0, so only the data from 15 to 60 days are shown in the figures. After 60 days of respective treatments, photographs were taken, and the number of shoots and collaterals were counted. After 90 days of respective treatments, shoot samples were taken to determine the contents of carbon- and nitrogen-metabolism-related products, lignin content, length, fresh weight, thickness, and puncture strength of shoots. Samples of shoots were also taken for phloroglucinol staining.

### 4.2. Determination of Shoot Length, Thickness, and Puncture Strength

Ten peripheral shoots with a height of 1.0–1.5 m and a consistent growth pattern were randomly selected around the canopy for marking, and the length of shoots was measured every 10 days from the base of the shoots to the growing point.

The thickness was measured with a vernier caliper at a distance of 1 cm from the base, and the length of the two internodes between the seedlings and new shoots was measured with a meter ruler, and the mean value was recorded as the length of the internodes (cm). The puncture strength was measured with a stem strength meter (DDY-1, Beijing Shunkeda Technology Co., Ltd., Beijing, China) at a distance of 10 cm from the base of the shoot.

### 4.3. Leaf Photosynthetic Rate Determination, Chlorophyll Content, and Stomata

At 15 and 60 days after respective treatments, the fifth fully expanded leaf on the upper part of the plant was selected, and the photosynthetic rate of the leaf was measured with a portable photosynthesis measuring system (PP Systems, Amesbury, MA, USA). Between 9:30 and 11:00 a.m., a SPAD-502 PLUS chlorophyll meter was used to measure the amount of chlorophyll in fully expanded leaves (Spectrum Technologies, Aurora, IL, USA). To perform the measurements of stomata, we stained the leaf epidermis with clear acrylic nail polish. The nail polish was taken off with forceps once it had dried. The solid polish was then placed on a microscope slide and examined with a fluorescence microscope at 400× magnification (AXI0, Carl Zeiss, Jena, Germany). We produced nine slides per treatment. We randomly selected three 1.4 × 10^3^ μm^2^ areas on each slide and took a picture. We used ImageJ version 1.48 to count the length and width of the stomata (National Institutes of Health, Bethesda, MD, USA).

### 4.4. Leaf Photosynthetic Rate Determination, Chlorophyll Content, and Stomata

The determination of lignin content was carried out with the plant lignin (lignin) enzyme-linked immunoassay (ELISA, Beijing Solarbio Science Technology Co., Ltd., Beijing, China) kit as per the manufacturer’s instructions and lignin content was determined using a microplate reader (Meigu Molecular Instruments Co., Ltd. Shanghai, China) at 450 nm.

For phloroglucinol staining, tissue sections were placed in 1% phosphomolybdic acid for 5 s, followed by rinsing in water and air-drying. Staining: tissue was surrounded, and hydrochloric acid (6 mol∙L^−1^) was then added drop-wise to the tissue until it was completely covered and allowed to stand for 30 s before an equal volume of 1% Phloroglucinol Stain A was added, then the resultant solution was mixed and allowed to stain for 2 min. After the removal of excess dye solution, the tissue sections were covered with a cover slip, and pictures were taken within 3 min to minimize dye fading. Because the material fades quickly, the photo should be taken quickly. The instruments and models used are shown in Table 5:

### 4.5. Soluble Sugar, Starch, Soluble Protein, Nitrate Nitrogen, and Ammonium Nitrogen Determinations

Soluble sugar contents were determined as described by Zhao and Cang [40]. Briefly, fresh leaves were washed, dried to completion, homogenized, and filtered through a 100 µm mesh sieve. Next, 0.1 g of the leaf powder was extracted in boiling water over 30 min. Aliquots of the extract were diluted 1:4 in anthrone reagent (add manufacturer here) and heated in boiling water for 10 min. After 10 min of cooling to room temperature, the soluble sugar content was determined for its absorbance at 620 nm (Ultraviolet-Vis Ultra-trace Spectrophotometer, Nanjing Feller Instrument Co., Ltd. Nanjing, China).

For the determination of starch contents, leaf powder was prepared and extracted in boiling water as described above, then the residue was boiled in 1.84 M perchloric acid for 15 min so that all starch was digested into glucose. The resultant sugar content was then determined using anthrone reagent as described above for total soluble sugars [40].

The soluble protein contents were determined from fresh samples as described by [40].

Nitrate nitrogen contents were determined from fresh samples as described by Zhao S and Cang J [40]. Ammonium nitrogen was determined as described in Zhao S and Cang J [40]. Determination of total nitrogen content in plants was performed by the Kjeldahl method (Haineng Future Technology Group Co., Ltd. Ji’nan, China).

### 4.6. Determination of Enzyme Activity Related to Carbon and Nitrogen Metabolism

NR activity was determined according to the method described in Zhao S and Cang J [40]. Fresh samples (0.5 g) were homogenized with 4 mL of 0.1 mol L^−1^ HEPES-KOH, pH 7.5, 3% (*w*/*v*) PVP, 1 mmol L^−1^ EDTA, and 7 mmol L^−1^ Cys. The assay mixture contained 50 mmol L^−1^ HEPES-KOH, pH 7.5, 100 mmol L^−1^ NADH, and 5 mmol L^−1^ KNO_3_ with 2 mmol L^−1^ EDTA or 6 mmol L^−1^ MgCl_2_. The assay volume was 2 mL. Activity was measured in crude extracts by determining NO_2_^−^ formation by the addition of 1% sulfanilamide and 0.2% N-(1-naphtyl)-ethylene-diamine dihydrochloride in 3 mol L^−1^ HCl. Activity state was defined by NR assay in the presence of Mg^2+^ (and 14-3-3 proteins) as a percentage of NR activity measured in the presence of EDTA, which reflected the quantity of the enzyme in the non-phosphorylated active form.

GS, GOGAT, SS, and SPS protein levels were determined using ELISA kits for plant GS, plant GOGAT, plant SS, and SPS (Kangwei Century Technology Co., Ltd., Beijing, China), respectively, as per the manufacturer’s instructions. The protein level of the commercial enzymes used in this study were determined based on the Folin method reported in Chinese Standard GB/T 23527-2009. Suitable diluted neutrase, protamex, and flavorzyme (1 mL) were separately added to 1 mL of 2% (*w*/*v*) casein in 50 mM phosphate buffer (pH 7.5), whereas the suitable diluted alcalase (1 mL) was added to 1 mL of 2% (*w*/*v*) casein in 50 mM borax–sodium hydroxide buffer (pH 10.5). The mixtures reacted at 40 °C for 10 min before stopping the reaction by adding 2 mL of 0.4 M trichloroacetic acid solution for 10 min. The mixtures were centrifuged at 10,500× *g* for 5 min at 4 °C, and then 1 mL of the supernatants was mixed with 5 mL of 0.4 M Na_2_CO_3_ and 1 mL of Folin–Ciocalteu reagent. The absorbance values were measured at 660 nm. One unit (U) of protease was defined as hydrolyze casein per minute to produce 1 μg tyrosine under assay conditions.

### 4.7. Membership Function Analysis

Membership function analysis [41] was performed on NR activity, SS, SPS, GOGAT, and GS protein level in leaves, phloem, and xylem of each treatment group by using the membership function method [41]. Membership function analysis was also performed on the contents of soluble sugar, starch, nitrate nitrogen, ammonium nitrogen, and soluble protein.

The membership function value of each index was obtained according to the following formula: Xu = (X − Xmin)/(Xmax − Xmin), where X is the measured value of a relevant parameter, and Xmax and Xmin are the maximum and minimum values of this parameter, respectively. The average value of the membership function of each parameter was then calculated. The larger the average value, the stronger the correlation is deemed to be.

### 4.8. Statistical Analysis

We collected three biological replicates for each treatment. Origin version 9.4 was used to conduct all statistical analyses. The data were subjected to one-way ANOVA and least significant difference test (Duncan’s new multiple range method, *p* < 0.05) using SPSS20.0 software. The threshold of statistical significance used for all tests was *p* < 0.05.

## 5. Conclusions

The application of valine can increase the thickness of shoots, lignin content, and puncture strength, and improve the degree of lignification of shoots. In conclusion, compared with urea, the application of valine can inhibit the growth of new shoots of peach trees, promote the accumulation of carbon and nitrogen nutrients in the plants, and improve the nutrient content and the degree of lignification in the plants; this is of great significance to improve the convenience of peach fruit harvesting.

## Figures and Tables

**Figure 1 plants-12-01596-f001:**
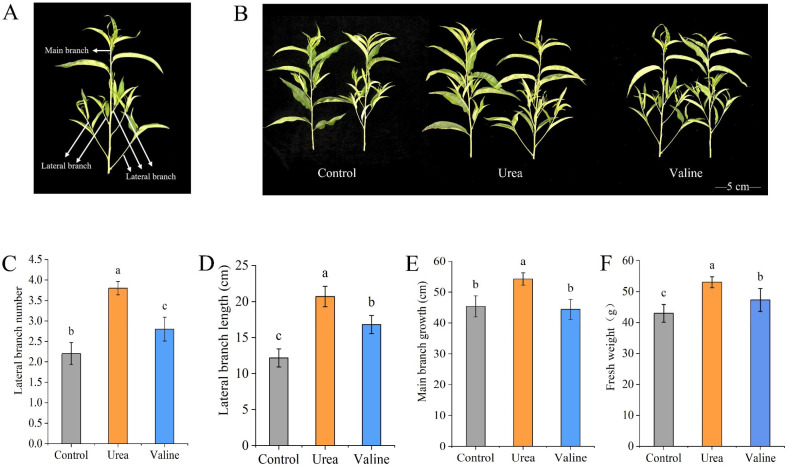
Effects of valine and urea on the growth of new shoots of peach trees: (**A**) Main branch and lateral branch distribution. (**B**) Growth of new shoots in different treatments. (**C**) Lateral branch number. (**D**) Lateral branch length. (**E**) Main branch growth. (**F**) Fresh weight. The error bar represents the standard deviation of the mean (*n* = 3). The experiment was conducted on 1 July 2020 using a valine and urea solution at a concentration of 4 g∙L^−^^1^. Different lowercase letters indicate significant differences among treatments (Duncan test, *p* < 0.05).

**Figure 2 plants-12-01596-f002:**
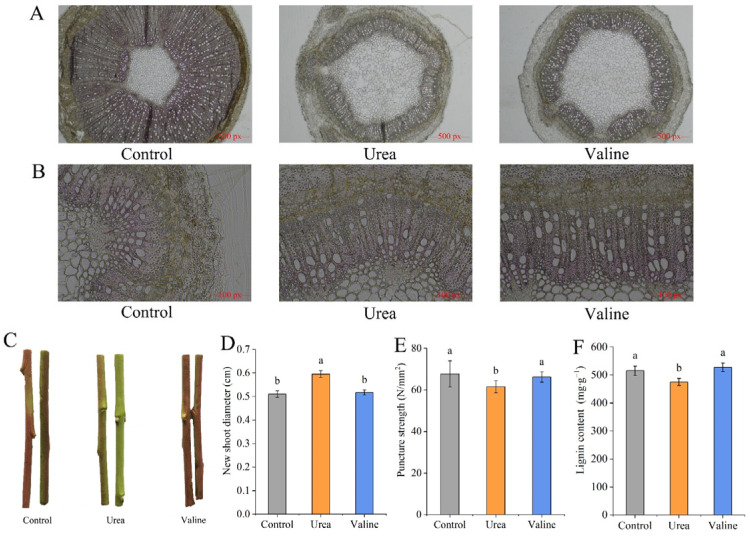
Effects of valine and urea on lignin content in new shoots of peach trees: (**A**) Effects of different treatments on phloroglucinol staining of peach new shoot; scale bar size is 500 px. (**B**) Effects of different treatments on phloroglucinol staining of peach new shoot; scale bar size is 100 px. (**C**) New shoot color. (**D**) New shoot diameter. (**E**) Puncture strength. (**F**) Lignin content. The error bar represents the standard deviation of the mean (*n* = 3). Different lowercase letters indicate significant differences among treatments (Duncan test, *p* < 0.05).

**Figure 3 plants-12-01596-f003:**
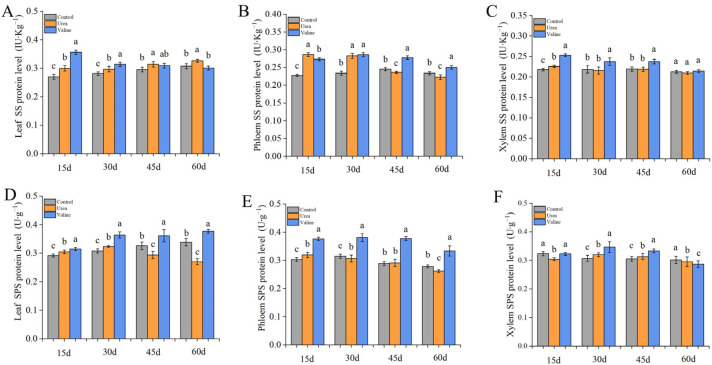
The effect of valine on carbon metabolism enzymes in peach tree. (**A**) Leaf SS protein level. (**B**) Phloem SS protein level. (**C**) Xylem SS protein level. (**D**) Leaf SPS protein level. (**E**) Phloem SPS protein level. (**F**) Xylem SPS protein level. The error bar represents the standard deviation of the mean (*n* = 3). Different lowercase letters indicate significant differences among treatments (Duncan test, *p* < 0.05).

**Figure 4 plants-12-01596-f004:**
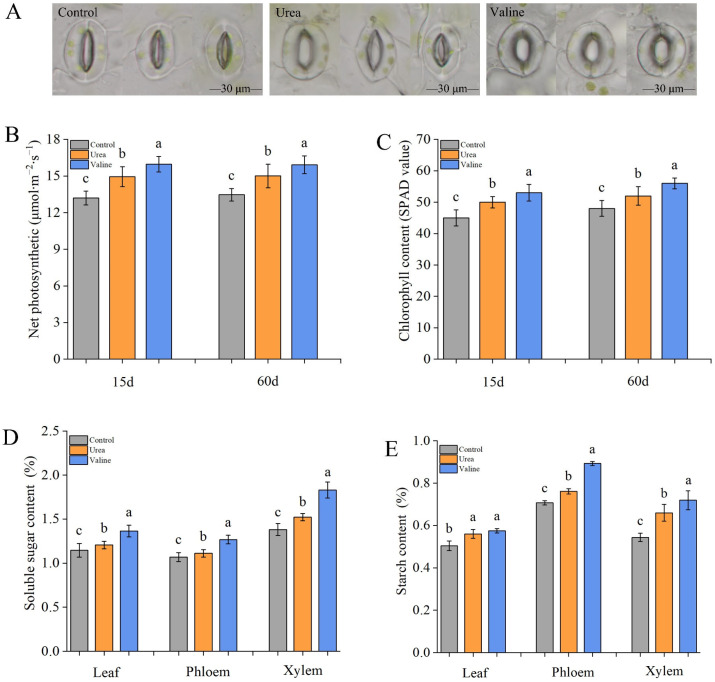
Effects of valine and urea on carbon metabolites in peach tree. (**A**) The shape and area of stomata. (**B**) Effects of different treatments on net photosynthetic rate of peach. (**C**) Chlorophyll content. (**D**) Soluble sugar content. (**E**) Starch content. Soluble sugar and starch contents were measured at day 60 after valine and urea treatment. The growth conditions, water status, and light conditions of the leaves used to collect stomata were the same. The error bar represents the standard deviation of the mean (*n* = 3). Different lowercase letters indicate significant differences among treatments (Duncan test, *p* < 0.05).

**Figure 5 plants-12-01596-f005:**
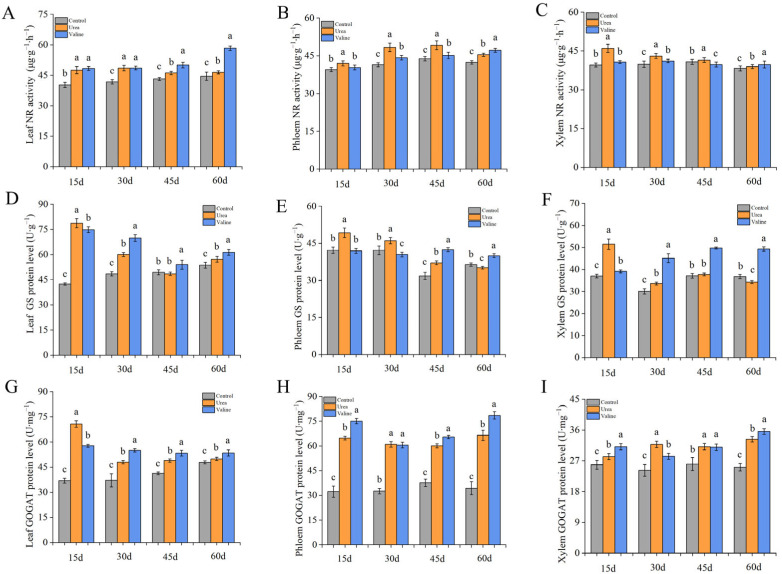
Effects of valine and urea on enzymes of nitrogen metabolism in peach tree. (**A**) Leaf NR activity. (**B**) Phloem NR activity. (**C**) Xylem NR activity. (**D**) Leaf GS protein level. (**E**) Phloem GS protein level. (**F**) Xylem GS protein level. (**G**) Leaf GOGAT protein level. (**H**) Phloem GOGAT protein level. (**I**) Xylem GOGAT protein level. The error bar represents the standard deviation of the mean (*n* = 3). Different lowercase letters indicate significant differences among treatments (Duncan test, *p* < 0.05).

**Figure 6 plants-12-01596-f006:**
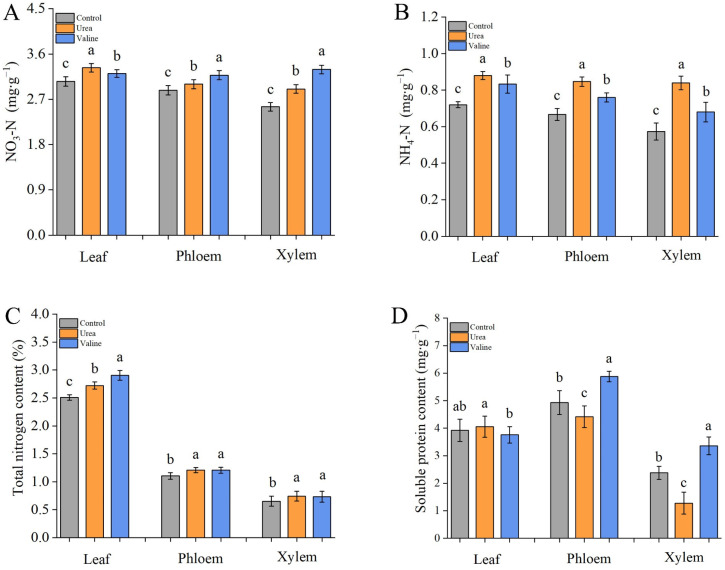
Effects of valine and urea on nitrogen metabolites in peach tree. (**A**) NO3-N content. (**B**) NH4-N content. (**C**) Total nitrogen content. (**D**) Soluble protein content. The error bar represents the standard deviation of the mean (*n* = 3). Samples were tested on day 60 after valine and urea treatment. Different lowercase letters indicate significant differences among treatments (Duncan test, *p* < 0.05).

**Figure 7 plants-12-01596-f007:**
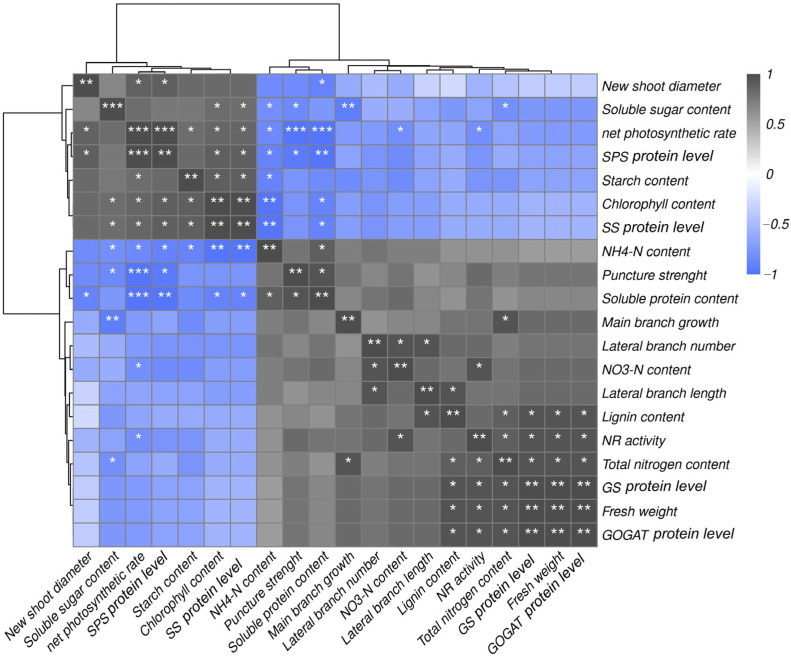
Correlation matrix among different physiological traits in peach tree. (Duncan test, * *p* ≤ 0.05, ** *p* ≤ 0.01, *** *p* ≤ 0.001)).

**Table 1 plants-12-01596-t001:** Stoma length, width, and area.

	*Prunus persica* (L.) Batsch. Stomatal Measurements		
	Length (μm)	Width (μm)	Area (μm^2^)
Control	27.4 ± 3.2 a	16.4 ± 1.1 c	319.1 ± 5.42 c
Urea	26.8 ± 2.4 b	18.3 ± 1.2 b	343.5 ± 6.71 b
Valine	26.3 ± 2.7 b	21.8 ± 1.5 a	415.3 ± 6.24 a

Note: Mean ± standard deviation (*n* = 3). The different letters indicate significant differences at a level of *p* < 0.05; the same below.

**Table 2 plants-12-01596-t002:** Comprehensive effects of the different treatments on key enzymes of carbon and nitrogen metabolism.

Treatments	Time	SS	SPS	NR	GS	GOGAT	Average
Leaf	Val	0.63	1.00	1.00	1.00	0.70	0.87
	Urea	0.45	0.00	0.51	0.77	0.67	0.48
	Control	0.00	0.32	0.00	0.00	0.00	0.06
Phloem	Val	1.00	1.00	0.33	0.81	1.00	0.83
	Urea	0.60	0.00	1.00	1.00	0.81	0.68
	Control	0.00	0.02	0.00	0.00	0.00	0.00
Xylem	Val	1.00	1.00	0.26	1.00	0.82	0.82
	Urea	0.03	0.06	1.00	0.38	0.83	0.46
	Control	0.00	0.15	0.00	0.00	0.00	0.03

**Table 3 plants-12-01596-t003:** Comprehensive effects of the different treatments on carbon and nitrogen accumulation.

Treatments	Soluble Sugar	Starch	NO_3_-N	NH_4_-N	Soluble Protein	Average
Val	0.71	0.21	0.81	0.52	1.00	0.65
Urea	0.20	0.32	1.00	1.00	0.00	0.50
Control	0.00	0.00	0.00	0.00	0.94	0.19

**Table 4 plants-12-01596-t004:** Effects of different concentrations of valine on the content of carbon and nitrogen metabolites and lignin.

Treatments	Concentration (g∙L^−1^)	Soluble Sugar Content (%)	Starch Content (%)	Total Nitrogen Content (%)	Lignin Content (mg∙g^−1^)
Val 1 g∙L^−1^	1	0.84 ± 0.09 e	0.45 ± 0.05 d	1.55 ± 0.09 e	473.2 ± 39.7 c
Val 2 g∙L^−1^	2	1.54 ± 0.12 c	0.66 ± 0.07 c	1.96 ± 0.15 d	507.4 ± 29.5 b
Val 4 g∙L^−1^	4	1.78 ± 0.11 a	0.85 ± 0.09 a	2.93 ± 0.16 a	532.7 ± 57.3 a
Val 8 g∙L^−1^	8	1.65 ± 0.07 b	0.76± 0.05 b	2.56 ± 0.13 b	470.5 ± 52.5 c
Val 16 g∙L^−1^	16	1.32 ± 0.07 d	0.43 ± 0.04 d	2.12 ± 0.17 c	455.3 ± 47.6 d

**Table 5 plants-12-01596-t005:** Instruments and models used in the experiment.

Name	Manufacturer	Model
Hard tissue slicer	Shanghai Leica Instrument Co., Ltd., Shanghai, China	HistoCore
Tissue Spreader	KEDEE, Guangzhou, China	Category 1
Upright Optical Microscope	Nikon Japan, Tokyo, Japan	Nikon Eclipse E100
imaging system	Nikon Japan, Tokyo, Japan	Nikon DS-U3

## Data Availability

Not applicable.

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
