# Peer review of "Effects of Valine and Urea on Carbon and Nitrogen Accumulation and Lignin Content in Peach Trees"

_plants, 2023, doi:10.3390/plants12081596_

Round 1

Reviewer 1 Report

Dear authors,

In this paper the authors study how the application of valine and urea influences different physiological parameters and the metabolism of carbon and nitrogen in peach trees. Determining that valine improves some of the physiological parameters studied.

The article has a series of important deficiencies that need its solution to verify the results obtained. If the authors can provide such data, I consider that the article would be publishable. If they cannot provide them, the hypotheses they try to prove would not be adequately tested.

Majors:

*L340: “Pre-experiment found that the valine concentration of 4 g·L−1 was the most effective, so it was set as experimental concentration.” This explanation is not sufficient to explain the importance of this fact. Since it is a critical point the concentrations of Valina and Urea used. The data carried out that lead to this conclusion should have been shown as a Fig. 1 of the paper. I do not understand how it is possible that both valine and urea show that in both the optimal concentration is resulted to be 4g/l; It seems to me too much of a coincidence, and therefore it should be tested experimentally and such results should be shown.

*In my opinion, the information does not provide certain necessary information and that should be provided. The role of lignin is not mentioned at all. The reason for the importance of lignin in this study should be explained in the introduction.

*The introduction does not mention the importance of Urea at all, an explanation of why it was used is necessary.

* In the introduction, a better explanation of why valine is used in particular is missing.

*L77:” PpSnRK1, PpTOR” define please

*In the legend of Fig.1, the concentrations of Valine and Urea used and the application time must be indicated.

*In the legend of Fig.2 there is an error, the F is not named and C, D and E are wrongly indicated

*L116: ”2.3 The effect of valine on carbon metabolism enzymes in peach tree” Why only valine? if you also study the effect of urea. The same L143.

*L144: “Figure 4A shows that the shape and size of stomata in peach leaves were different

after urea and valine treatment”  I don't understand why these data are put in this section? What do they have to do with the other parameters measured in it? please explain.

*Fig 4A does not have to do with carbon metabolites, as the legend says, remove or change the legend. Change the legend to Fig 4, indicating that urea was also used.

*In table 2 there is a number 168? what's that

*Legend to Fig.4D y E. Indicate the time in which the samples were taken

*Legend to Fig.6. Indicate the time in which the samples were taken.

* In the discussion, the middle of the first paragraph would fit better in the introduction.

*L295-L299: It is needed to indicate the references

*In the discussion it is necessary to refer to and discuss previous works that have used valine or other amino acids in similar studies.

Minors:

*L245: The the net. Typo

Reviewer 2 Report

In this manuscript, the authors compare the effects of urea and valine as fertilizers for peach tree growth. Effects on shoot growth, lignin contents, and some aspects of carbon and nitrogen metabolism were evaluated. English-language editing and extensive revision of the analysis are necessary, with particular attention to the following points:

Line 75 – This states why valine was selected, but the introduction should also include a statement of why urea was used for comparison.

Line 88 – No data showing the differences in leaf area are given to support this statement. There is also no mention of how leaf area was measured in the Materials and Methods.

Lines 89-92 – There is no mention of how urea- and valine-treated plants compared to the control. As the growth of valine-treated plants is more similar than that of urea-treated plants to the control, this comparison would be useful for the argument that valine is a preferable nitrogen source.

Line 92 – If different lowercase letters indicate significant differences among treatments, then Fig. 1F shows that valine-treated plants had significantly lower biomass fresh weight than urea-treated plants. The statement that the reduction was insignificant is incorrect.

Line 101 – Figure 2A does not appear to show higher lignin content in valine-treated shoots than in the control, particularly as the lignified layer is much thicker in the control than in valine-treated shoots. Indeed, Figure 2F directly contradicts the statement that valine-treated shoots had the highest lignin content of the three groups, as it shows the lignin contents of control and valine-treated shoots to be similar.

Lines 102-109 and Figure 2 caption – The panels of the figure do not match their descriptions in the text or caption. Either the panel A or the panel B shown in the figure is not captioned or described in the text.

Lines 125-127 – It would improve the argument that valine is a preferable fertilizer if it were mentioned that SPS activity in valine-treated shoots is significantly greater than that in urea-treated shoots in every tissue and timepoint tested except one.

Lines 144-147, Figure 4A, and Table 2 – It appears in Figure 4A that the stomata of the valine-treated plants are just more open than those in the other two groups. As there are many factors that influence stomatal opening, how can it be determined that the differences observed are due to the fertilizer? For example, were the water statuses and light conditions of all plants the same when the stomatal peels were taken?

Lines 152-153 – If different lowercase letters indicate significant differences among treatments, the statement that urea did not affect soluble sugar content is incorrect.

Lines 145-155 – Again, the statement in the text does not match what is shown in the figure. Figure 4E clearly shows an increase in leaf starch content in urea- and valine-treated leaves.

Line 181 – I believe this should specify the GS enzyme activity of “valine-treated” leaves.

Section 2.6 – No timepoint is given. When were these samples collected?

Lines 292-293 – Premature senescence was not previously mentioned. Would this have any impact on fruit production, thereby being a drawback of urea or nitrate treatment?

Round 2

Reviewer 1 Report

I believe that the authors have responded favorably to most of my suggestions, I accept the paper in its current version.

Author Response

We appreciate for Editors and Reviewers’ diligentwork and inspiring opinions earnestly. Once again, thank you very much for your comments and suggestions.

Reviewer 2 Report

The authors have responded adequately to most of my concerns. In line 103, "insignificant" should be replaced by "significant" for the text to agree with what is shown in the figure.

Author Response

Thanks for your valuable comments. Considering the Reviewer’s suggestion, we have modified it to 'significant'. (Line 103)